# Modelling ribosome kinetics and translational control on dynamic mRNA

**Eric C. Dykeman**⬦*¤

Department of Mathematics, University of York, York, United Kingdom

¤ Current address: Department of Mathematics, University of York, York, United Kingdom
* eric.dykeman@york.ac.uk

## Abstract

The control of protein synthesis and the overall levels of various proteins in the cell is critical for achieving homoeostasis. Regulation of protein levels can occur at the transcriptional level, where the total number of messenger RNAs in the overall transcriptome are controlled, or at the translational level, where interactions of proteins and ribosomes with the messenger RNA determine protein translational efficiency. Although transcriptional control of mRNA levels is the most commonly used regulatory control mechanism in cells, positive-sense single-stranded RNA viruses often utilise translational control mechanisms to regulate their proteins in the host cell. Here I detail a computational method for stochastically simulating protein synthesis on a dynamic messenger RNA using the Gillespie algorithm, where the mRNA is allowed to co-translationally fold in response to ribosome movement. Applying the model to the test case of the bacteriophage MS2 virus, I show that the models ability to accurately reproduce experimental measurements of coat protein production and translational repression of the viral RNA dependant RNA polymerase at high coat protein concentrations. The computational techniques reported here open up the potential to examine the infection dynamics of a ssRNA virus in a host cell at the level of the genomic RNA, as well as examine general translation control mechanisms present in polycistronic mRNAs.

## Author summary

The regulation of the proteome in the cell occurs via two main mechanisms, transcriptional control of mRNA quantities in the cell, or via translational control, where ribosome and protein interactions with the mRNA determine protein translational efficiency. Examples of translational control often occur in positive-sense single-stranded RNA viruses, where interactions with secondary structures in the viral mRNA regulate the levels of viral proteins in the host cell. Understanding translational control on individual mRNAs thus requires examining the dynamics of mRNA folding in response to ribosome translocations. Here I describe a stochastic model based on the Gillespie algorithm which is capable of simulating ribosome kinetics on dynamic mRNAs which co-translationally fold with ribosome movements on the mRNA. The resulting model is applied to a case study of the bacteriophage MS2, where mechanisms such as translational coupling and

**Data Availability Statement:** All relevant data are within the manuscript and its Supporting information files. Computational code can be downloaded from the Authors webpage at https://www-users.york.ac.uk/~ecd502 or from Github at https://github.com/edykeman/ribofold.

**Funding:** The Author has received no funding for this work.

**Competing interests:** The Author declares no competing interests.

translational repression regulate quantities of coat and RNA dependant RNA polymerase in the cell.

## Introduction

The regulation of cellular processes, in particular protein synthesis and the composition of the proteome, is a multi-factor process arising from complex feedback and regulatory systems in the cell. In the simplest of terms, the regulation of the proteome occurs via two basic routes, (1) at the transcriptional level, where the amount of protein in the cell can be controlled via the overall amount of messenger RNA (mRNA) in the cell or (2) at the translational level, where the amount of protein produced from a single mRNA is controlled via interactions of the mRNA with other proteins and the ribosome itself. While the former control mechanism requires understanding the process of transcription by RNA polymerase along with its regulation and feedback mechanisms in order to predict its impact on protein levels in the cell, the latter requires examining the structure of the mRNA and its co-translational folding in response to ribosome movement.

Examples of translational control and regulation of protein synthesis via mRNA secondary structure are frequently found in positive-sense single-stranded RNA ((+)ssRNA) viruses. Since (+)ssRNA viruses lack a DNA stage to their infection cycle, they rely solely on the genomic RNA, which acts directly as a messenger RNA in the cell, along with any sub-genomic fragments produced, to regulate viral protein levels during the infection cycle. Translational regulation of protein synthesis in these viruses has been noted to occur by mechanisms such as: (1) ribosome interaction with secondary/tertiary structure of the mRNA (e.g. Internal Ribosomal Entry Sites—IRES [1]), (2) translational coupling, where the translation of downstream genes are dependent on translation of upstream genes (e.g. bacteriophage MS2 [2]), or (3) via interactions with viral and host proteins with mRNA which repress or promote translation (e.g. translational repression in bacteriophage MS2 [2]), or promote frame-shifting events (+1 frame-shifting by viral 2A protein in cardiovirus [3]). In each of these examples, regulation of protein synthesis occurs via interaction of viral proteins or host ribosomes with specific secondary structures in the mRNA, or is the result of structural re-modelling of the mRNA in response to ribosome movement. Bacteriophage MS2 serves as an example of the latter case, where the movement of the ribosome through the coat gene re-models the structure of the viral mRNA by disrupting long distance RNA base-pairs [2, 4]. This results in the exposure of the translation initiation region (TIR) of the downstream RNA dependant RNA polymerase (RdRp) gene, allowing host ribosomes access to this gene which was previously hidden by the secondary structure of the mRNA [2, 4].

In addition to (+)ssRNA viruses, there is some suggestion that bacterial DNA viruses, such as bacteriophage P22, also use translational control and translational coupling mechanisms in their mRNA to regulate the ratios of viral structural proteins required for the efficient production of viral capsids. Although specific translational control mechanisms have yet to be elucidated, the structural proteins for P22 are synthesised from a single 20kb polycistronic mRNA [4] which contains the genes for both scaffolding and capsid protein. Experiments from the Teschke lab [5] have shown that sub-stoichiometric ratios of scaffolding/capsid protein are required for efficient assembly of pro-capsids, hence suggesting some form of translational control is present as both genes are expressed on a single mRNA.

Aside from viral systems, additional examples of translational control have also been observed in bacteria, in particular the regulation of large ribosomal protein expression. For

example, the IF3-L35-L20 polycistronic mRNA in *E. coli* is regulated both by translational coupling and translational repression via negative feedback from L20 concentrations in the cell [6]. Similar translational coupling and repression has also been observed in the L11-L1 mRNA in *E. coli* [7]. In both of these examples, translational coupling between upstream and downstream genes insures a ratio of protein expression of 1:1, enforcing the required ratio of large protein subunits for assembly of mature ribosomes [8]. Finally, while polycistronic mRNAs are commonly observed in prokaryotic cells and (+)ssRNA viruses infecting Humans, animals, and plants, recent work has highlighted examples of polycistronic mRNAs in mammalian genes, where expression is controlled via IRES elements [9]. This suggests that translational control is a universal phenomenon, with evidence for existence in both eukaryotic and prokaryotic mRNAs.

A detailed look at all of the above translational control mechanisms observed across bacteria and viruses illustrate the importance of accounting for the secondary structure of the mRNA, its interactions with proteins and ribosomes, and the co-translational folding of mRNA in response to ribosome movement when making predictions on the amount of protein expressed from an mRNA containing multiple genes. Motivated by understanding the regulatory control of mRNAs at the translational level, this paper details my development of a kinetic model of the ribosome which incorporates co-translational folding of the mRNA in response to ribosome movement. The new model reported here extends my previous stochastic model for studying *in vivo* ribosome kinetics [10] through the incorporation of kinetic reactions for the co-translational folding of the mRNA, along with additional reactions allowing proteins to bind to the mRNA at specific structural sites. Since my previous model simulated ribosome movements on full-length mRNAs with explicit nucleotide information, the level of detail on the RNA structure in this model is also at the single nucleotide level, enabling detailed questions to be probed theoretically. One such example is how nucleotide mutations to an mRNA affect the expression of protein. Since my model accounts for the resulting changes to mRNA dynamics that would occur from a mutation, it is able to examine such questions. Moreover, while previous models have examined the role of RNA structure in the initiation process and its effects on gene expression [11, 12], the RNA folding reactions considered in my model here takes into account structural changes in the whole of the mRNA due to ribosome movement, allowing for a detailed examination of features such as translational coupling and its impact on gene expression.

Using the bacteriophage MS2 system as an example, I demonstrate how my ribosome model with co-translational folding is able to reproduce the experimentally observed translational coupling between viral coat protein and RdRp genes, as well as the observed translational repression of RdRp that occurs as coat protein levels increase. I compare the predicted protein expression with relative coat protein expression ratios that were measured experimentally for a variety of phage mutants [13] and show that my model is in close agreement. Finally, a comparison of the simulation results with these experimental measurements allows me to determine estimates for several kinetic parameters for ribosome initiation, as well as the kinetics of RNA folding.

## Results

### Stochastic model of ribosome kinetics on a dynamic mRNA

In my previous work [10], I developed a stochastic ribosome kinetics model which simulates the total protein synthesis that occurs in an entire prokaryotic cell such as *E. coli*. While many protein synthesis models typically examine the translational dynamics on a single mRNA of interest, there are several advantages in taking into account the translational dynamics of the

full transcriptome. First, it allows transcriptome dependent effects which may have downstream impacts on the translation of the gene of interest, such as tRNA usage, to be accounted for. Second, it allows more accurate estimates of the free concentrations of various proteins involved in the translational process, such as free Ef-Tu and free 30S and 50S subunits, to be determined. For example, Ef-Tu forms a complex with Ef-Ts that is independent of the translational machinery but important for the re-charging process. Thus, the amount of Ef-Tu that is free and available to bind to translating ribsomes depends on these re-charging reactions and illustrates the importance of considering a holistic model of translation. While my previous model accounts for the individual nucleotide sequence of each mRNA that is present in the transcriptome of the cell, there is no mRNA structure, and each mRNA is treated as an unstructured linear sequence. Furthermore, each mRNA is treated as monocistronic and there is a single binding rate for the 30S binding to mRNA which is one of the main kinetic reactions that governs the initiation rate of protein synthesis. Extending this model to take into account the polycistronic nature of some mRNAs, as well as both the secondary structure of the mRNA and its folding kinetics, requires incorporation of specific computational algorithms. Specifically, dynamic programming algorithms to predict the secondary structure of the mRNA [14], as well as specialised data structures to efficiently store information on each ribosome position on the mRNA along with the structural features of the mRNA that occur around bound ribosomes. Incorporation of such secondary structural features are important for identifying downstream hairpins which are part of TIR regions or promote frame-shifting events. It should be noted that pseudo-knots are not considered in the extended ribosome model as these are difficult to predict using dynamic programming algorithms. However, there may be scope to include simple pseudo-knots in the future.

Since secondary structure prediction algorithms typically require the pre-computation of an $O(n^2)$ matrix containing the information on lowest energy structure on every RNA fragment $(i, j)$ with $i, j \in [1, n]$, it is not yet practical to consider the structure of *all* possible mRNAs in the transcriptome simultaneously. Instead, I consider the secondary structure and corresponding dynamics for a single mRNA "type", which could be present multiple times in the transcriptome. This enables the dynamics of mRNA folding in response to ribosome movements and protein binding to be studied for a single mRNA of interest (which I will refer to as the dynamic mRNA) while the background of ribosome activity of cellular mRNAs (which I will term background mRNAs) are treated as unstructured linear mRNAs as in my previous model. Examples of mRNAs that could be studied are viral mRNAs, or alternatively one could choose a specific gene operon of interest, e.g. the IF3-L35-L20 mRNA in *E. coli*, and theoretically probe its translational control while in the background of ribosome activity on other cellular mRNAs in the transcriptome.

In addition to incorporating the dynamics of mRNA folding in response to ribosome movements, I have also included the ability of proteins, either present in the cell or produced from the dynamic mRNA, to bind to secondary structural elements in the dynamic mRNA (either hairpins or multi-loop helices). Moreover, TIRs in the dynamic mRNA are also identified in the model based on the surrounding mRNA secondary structure. The 30S binding rate to the TIR is then predicted following a slightly modified kinetic version of the Salis model [11] (see Supporting information), which takes into account mRNA secondary structure around the TIR and the time required for the mRNA to melt and expose the start codon. This feature allows for polycistronic mRNAs to be considered, and for translational coupling, where mRNA structure hides downstream ribosome binding sites, to be incorporated. It also allows the consideration of dynamic changes to the initiation rate due to ribosomes or other proteins altering the structure of the TIR. Borujeni and Salis have noted that such situations occur during ribosomal drafting, where ribosome movement over the TIR prevents re-folding of the

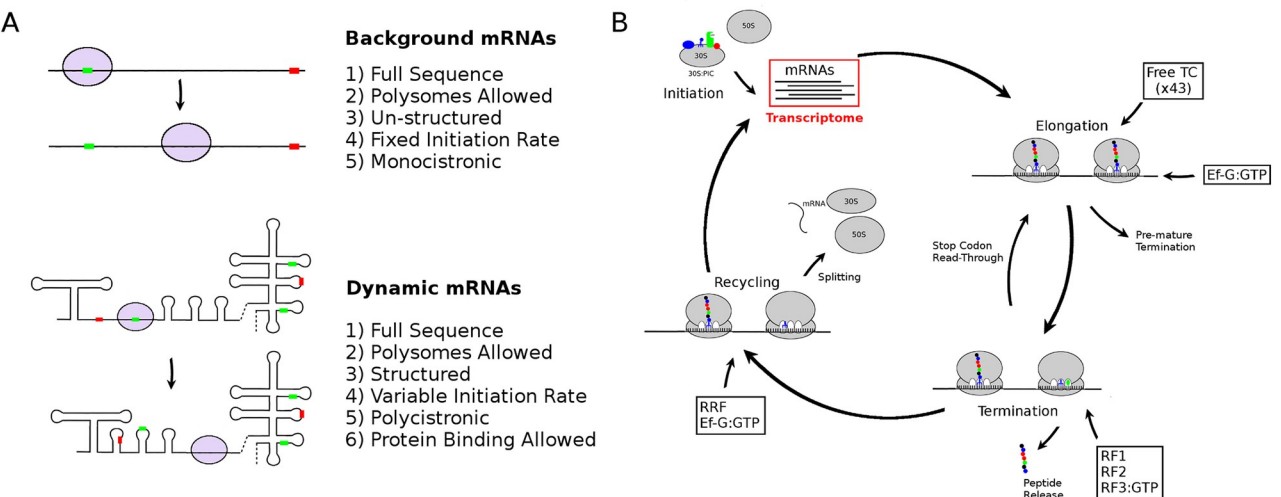

**Fig 1. A model of ribosome kinetics on dynamic mRNAs.** (A) The transcriptome in the model consists of two types of mRNAs: background mRNAs which are considered as monocistronic with no secondary structure, and dynamic mRNAs which are polycistronic with secondary structure. Initiation rates at potential start codons (green bars, with red bars indicating stop codons) can vary in dynamic mRNAs depending on the current structure around the translation initiation region (TIR) while they are modelled as fixed in the background mRNAs. The movement of the ribosome during elongation is illustrated. (B) The transcriptome is modelled using the framework of the stochastic model in [10], with added RNA folding reactions on the dynamic mRNA.

TIR, thereby allowing subsequent ribosomes easier access to the initiation site [15]. Fig 1 summarises these features and shows how my previous stochastic model has been extended to incorporate a dynamic mRNA. Fig 1A shows the two different types of mRNAs considered in the extended model, while Fig 1B illustrates the general model used to simulate ribosome kinetics on both structured and un-structured mRNAs. The individual reaction steps considered in the model cover all known kinetic steps of ribosome initiation, elongation, and termination. For example, the ribosome elongation step in the model encompasses 9 individual kinetic steps covering concentration dependant recruitment of ternary complex, GTP hydrolysis, and EF-G dependant translocation of the ribosome. Such reaction steps have been also been accounted for in similar models of translation [16], including the dependence on tRNA concentrations [17]. Similar kinetic detail is also present in the model for the initiation and recycling/termination stages, while the additional reactions involving re-charging of ternary complex and GTP/GDP exchange on GTPases are also accounted for. For full details of the individual kinetic steps see [10].

The following sections give a brief overview of the methods/algorithms used to; (1) incorporate co-translational folding (2) identify ribosome binding sites, and (3) include reactions for protein binding to mRNA structures, using the mRNA from bacteriophage MS2 coat and RNA dependant RNA polymerase (RdRp) genes as an example. Technical information on the implementation of specific algorithms and data structures are discussed in detail in the supporting information.

**Incorporating mRNA secondary structure and co-translational RNA folding kinetics.** To incorporate the structure of the mRNA and its kinetics into my previous stochastic ribosome kinetics model [10], I have developed a specialised tree representation of the mRNA structure that allows for (1) a finite set of folding transitions between mRNA states to be computed efficiently and (2) for the kinetic rates of these folding transitions to be stored in a separate binary tree. The set of mRNA states accessible in the model are obtained by computing

locally optimal structures in fragments between nucleotides $i$, $j$ in the mRNA, similar to the procedure in Geis et al [18]. The kinetic rates between these mRNA states are then determined using a breadth-first-search path finding algorithm [19, 20] which identifies the path with lowest energy barrier using Turner 99 rules for the base-pair energies [21]. This results in a co-translational RNA folding algorithm where the state space for the mRNA is coarse-grained, but kinetic rates are estimated from transition paths at single nucleotide resolution. Full details of the computation of kinetic rates for folding trajectories from minimum energy barrier pathways, as well as how a finite set of folding transitions are constructed using the tree-representation of the mRNA can be found in the supporting information. Fig 2 shows an example of the tree representation for a section of the bacteriophage MS2 mRNA encoding the coat protein. It is important to note that, while the tree representation encodes the structure of the mRNA in a coarse-grained manner allowing quick identification of local structures, the full secondary structure of the mRNA is also stored and adjusted in response to folding reactions and ribosome movements.

Regarding the storage of the kinetic rates on a binary tree, this is important for the computational efficiency of the algorithm as it allows a single Gillespie step to be implemented in logarithmic time. Here, the binary tree enables one to find and choose a folding reaction to fire in $O(\log_2(m))$ time, where $m$ is the number of structural helices (either a hairpin or long-distance interaction) present in the mRNA. In my previous ribosome kinetics model, the overall run time for a single Gillespie step (the computational time to "fire" a reaction, update the reaction list, and re-sum all of the reaction propensities) scaled with the number of mRNAs $N$ as $O(\log_2(N))$. Thus, the expected run time for a single Gillespie step on a set of $N$ dynamic mRNAs allowed to co-translationally fold is expected to be roughly $O(\log_2(mN))$. Comparing run times with and without dynamic mRNAs, I have found that the computational cost of simulating dynamic mRNAs with lengths $\approx 1.5$ to 3.5k nucleotides is roughly similar to that of my previous ribosome kinetic model, which was able to simulate roughly 30 min of protein synthesis that occurs in a cell in approximately 10 CPU hours.

**Prediction of translation initiation regions and ribosome binding rates.** To study prokaryotic ribosome initiation on a polycistronic mRNA, one requires the location of the appropriate translational initiation region (TIR) for the ribosome and corresponding start codon. Moreover, in order to accurately estimate the protein expression, the kinetic rate of the 30S ribosome subunit binding to the mRNA and initiating at the start codon are also required. On the one hand, the information on start codon positions can be obtained from an analysis of continuous coding regions in the mRNA that are not interrupted by stop codons. Using this method, one can construct a fixed list of start positions and TIRs. However in reality, the prokaryotic ribosome knows none of this information and only operates by simply attempting to bind to predominantly single-stranded regions of the mRNA [22, 23]. Thus, successful initiation of the ribosome requires; (1) a favourable interaction with a start codon (AUG or GUG preferably) in the mRNA with the 30S pre-initiation complex (30S:PIC) and (2) a predominantly single stranded region with relativity weak secondary structure to be available (i.e. free of translating ribosomes and protein). Thus, TIRs can potentially be blocked by translating ribosomes and/or by formation of strong secondary structure elements. Moreover, TIRs are potentially dynamic in nature, being sequestered in secondary structure or exposed in response to ribosome movements, protein binding, and mRNA folding. Thus, I have adopted a more general algorithm for the prediction of TIRs as opposed to simple identification of open reading frames using bio-informatics methods, in addition to allowing users to specify specific fixed start codons in the computational code.

For the prediction of possible TIRs in the mRNA, and the corresponding kinetic binding rate of the ribosome to the TIR, I use the following algorithm. For a deeper discussion of the

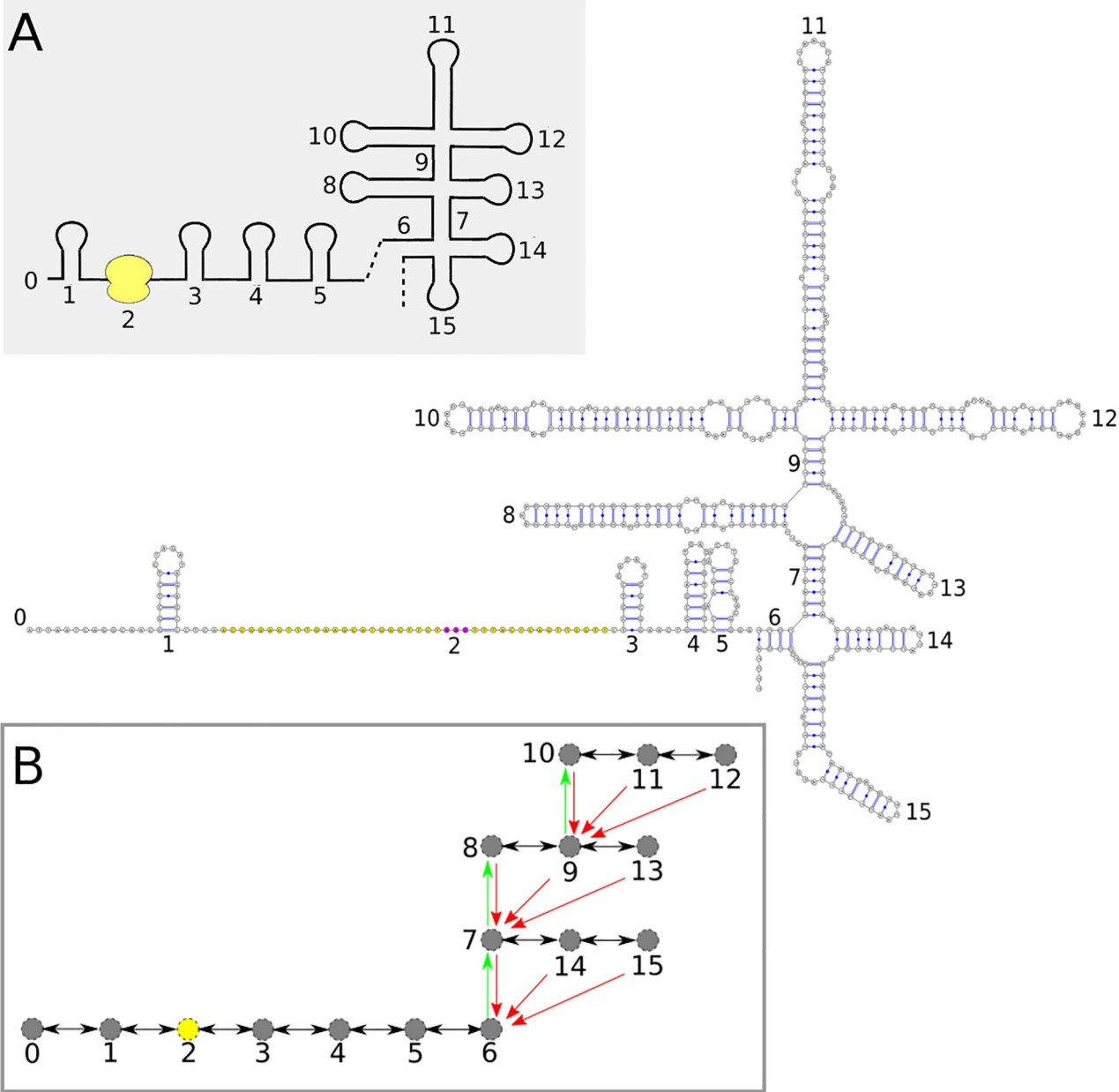

**Fig 2. Tree-representation of the bacteriophage MS2 coat protein gene.** (A) The inset gives a coarse-grained cartoon diagram of the full-nucleotide structure with explicit base-pairing, which is stored in the model. Bound ribosome is coloured yellow, with purple bases showing the location of the ribosome P-site. (B) The coarse-grained representation of the mRNA structure is translated into the rooted tree data structure shown, where each numbered node in the tree represents a coarse-grained helix in the cartoon diagram. Green arrows denote links to leaf nodes, while red arrows denote links to root nodes. Links to the main root node 0 denoting the 5' end of the mRNA are not shown for simplicity. Black arrows between nodes show links in the linked list data structure which stores neighbour information for the tree.

technical details of the algorithm, please see the Supporting information. First, the linear sequence of the mRNA is examined for potential start codons (only AUG,GUG,CUG,UUG, AUA,AUC,AUU are considered) and the optimal interaction of the 30S subunit with any upstream Shine-Dalgarno sequence is estimated. This gives a list of potential TIRs, each with a corresponding energy of interaction with the 30S subunit $\Delta G_{30S:mRNA}$ (see Supporting

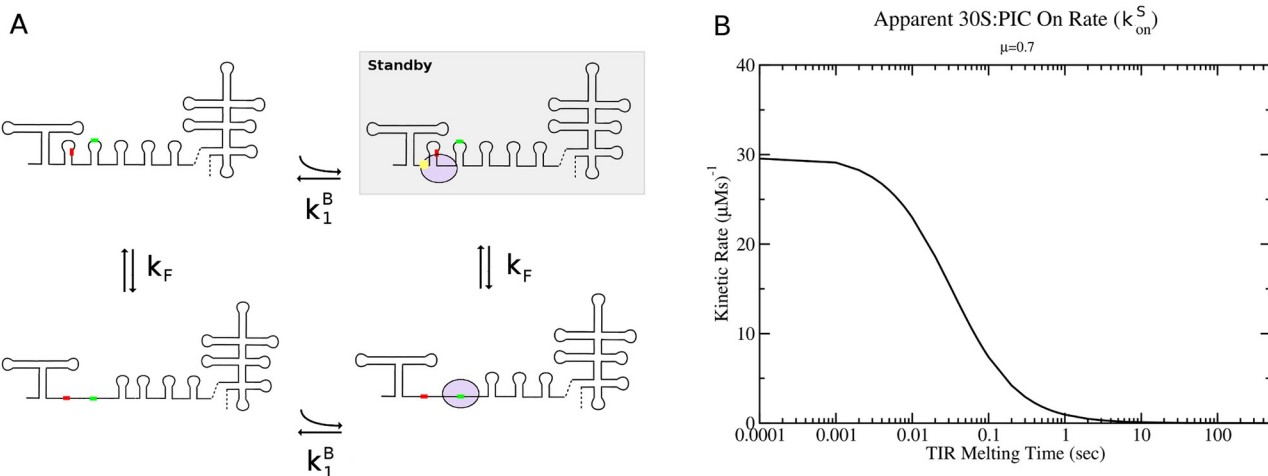

**Fig 3. Model of ribosome binding kinetics to a translation initiation region.** (A) Kinetic model of 30S:PIC binding to a TIR region of an mRNA via standby and direct pathways. TIR regions are considered as being hairpin only regions, possibly flanked by multi-loop helices. $k_{-F} = 1/\tau_u$ is the kinetic rate of TIR unfolding, while $k_1^B$ is the binding rate of 30S:PIC to the mRNA via ribosome protein S1. (B) Apparent 30S:PIC binding rate to TIR region at a cellular growth rate of $\mu = 0.7$ doublings per hour.

information for details on how this energy is calculated). The list is pruned to remove overlapping TIRs by keeping the TIR with lowest $\Delta G_{30S:mRNA}$. This assumes that 30S subunits will thermodynamically equilibrate on the lowest energy TIR in the region, which should be a good approximation for standby sites. The resulting list then encompasses all potential non-overlapping TIR sites in which the 30S subunit could bind and attempt to initiate translation at and this list fulfils criterion (1) discussed above. Successful initiation at one of these TIRs requires fulfilling criterion (2), i.e. that the TIR to be located in a region of the mRNA with weak secondary structure. As the mRNA is to be considered dynamic, the calculation of the kinetic binding rate of the 30S:PIC to the TIR must come in response to co-translational folding and/or melting of the mRNA. Van Duin and other colleagues have suggested that the 30S:PIC binds to these weak regions via the small ribosome protein S1, which binds non-sequence specifically to single-stranded RNA regions of 5–20 nucleotides [23, 24]. Thus I consider a "weakly structured" region to be a hairpin only region of the mRNA, potentially flanked by two multi-loop helices (c.f. Fig 3A), encompassing at least 20 single stranded nucleotides following analysis from Van Duin et al. [25]. While the model and computational code does allow for initiation at non-canonical start codons differing from AUG by one nucleotide, it should be noted that detailed experimental information on how the kinetic rates governing 50S recruitment are effected by non-canonical start codons is not currently available. Hence the model is likely un-reliable in estimating initiation frequency at non-canonical starts at the present time.

To calculate the kinetics of 30S:PIC binding to the TIR, I use the model illustrated in Fig 3A (c.f. detailed Fig H in S1 Text). I assume that the ribosome first interacts non-sequence specifically with a weakly structured area of the mRNA via small ribosomal protein S1. If the TIR is sequestered in secondary structure, the ribosome waits as a standby ribosome for the TIR to unfold [24]. After unfolding of the TIR, the ribosome is free to engage with the start codon and for the anti-SD section of the 16S rRNA to interact with any upstream Shine-Dalgarno sequence present in the mRNA. Together, these sets of kinetic steps govern the apparent binding rate of the 30S:PIC to the TIR of the mRNA, along with the apparent off rate. One can derive an explicit formula for the apparent on rate for 30S:PIC binding to the TIR region

containing a standby site from a calculation for the mean-first-passage time, i.e. the average time it takes for the 30S subunit to bind the TIR and engage with the start codon. From Fig 3A (c.f. Fig H in S1 Text) the mean-first-passage time for 30S binding to a start codon via the standby site pathway ($\tau_{on}^s$) is given by

$$
\begin{aligned}
\tau_{on}^s &= \frac{1}{r_f} + \frac{k_{-1}^B}{r_f}\frac{1}{k_{-F}} + \frac{1}{k_{-F}} \\
&= \frac{k_{-F} + k_{-1}^B + r_f}{r_f k_{-F}}.
\end{aligned}
$$

Here, the value of $r_f = k_1^B[30S]$ is the rate of S1 binding to mRNA times the concentration of free 30S:PIC subunits. Using $k_{on}^s[30S] \approx 1/\tau_{on}^s$ along with the average time for TIR unfolding, $\tau_u = 1/k_{-F}$ we can estimate the apparent 30S binding rates as

$$
k_{on}^s = \frac{k_1^B}{1 + \tau_u(k_1^B[30S] + k_{-1}^B)} \tag{1}
$$

where $k_1^B = 30\mu M^{-1}s^{-1}$ and $k_{-1}^B = 10s^{-1}$ are the best fit kinetic parameters for the on and off rates of 30S:PIC subunit binding to the mRNA via ribosomal protein S1. The parameter $[30S]$ is the concentration of free 30S:PIC subunits in the cell while $\tau_u$ is the TIR unfolding time. A similar formula for the apparent off rate is derived in the supporting information. At a growth rate of $\mu = 0.7$ doublings per hour, my kinetic ribosome model estimates that the free concentration of 30S:PIC subunits is roughly 0.65 $\mu M$. This value is used to generate the plot in Fig 3B, which shows the apparent on rate of 30S:PIC binding via a standby site to the TIR region as a function of TIR unfolding time $\tau_u$.

This model of initiation differs from that of Salis *et al.* [11] in that it incorporates the effect of S1 binding and is kinetic rather than thermodynamic. One advantage of this is that, for mRNA sequences which lack an SD sequence, the binding affinity and resulting off rate of the 30S subunit is controlled by the interaction of S1 with mRNA. Von Hippel and Draper [23] have measured the association constant to be roughly $K_a = 3 \times 10^6 M^{-1}$ which gives an interaction energy of $\Delta G_{S1} = -9.19$ kcal/mol, and this is weaker then the expected interaction energy of anti-SD sequence with a complementary SD ($\Delta G_{SD} = -12.1$ kcal/mol). Thus, this model forces the binding affinity of 30S subunit to range between these two values, and the resulting apparent kinetic off-rates for the 30S subunit can be easily parameterised to lie between $k_{off} \in$ [0.001, 10.0] per second, consistent with the order of magnitude for values of $k_{off}$ measured by Studer and Joseph [22], who measured 30S subunit binding to mRNAs with and without complementary SD sequences. Their measurements give values for the two extremes, $k_{off} = 0.001 s^{-1}$ for sequences containing a full complementary SD sequence, and $k_{off} = 4s^{-1}$ when the SD sequence is not present. A full technical discussion and derivation of the kinetic rates can be found in the supporting information.

**Modelling the binding of proteins to structures in the mRNA.** Modelling the binding of protein to specific secondary structures in the mRNA is also accounted for in this model. Each node in the tree corresponds to a helical region of the mRNA (either hairpin or multi-loop) and the computational code allows one to model the binding of proteins to these helical sections of the mRNA. Proteins are allowed to bind at any helix, thus for the case of the MS2 mRNA, coat proteins will bind at any hairpin containing the correct binding motif, not just the TR hairpin. This is important since MS2 is known to have additional coat protein binding sites important for viral packaging and assembly. As the exact nucleotide and base-pairing arrangement is also stored, a detailed sequence dependant binding profile for the protein is

easily implemented in the model. Similarly, for proteins which bind single-stranded regions, this can be modelled as binding to pairs of neighbouring nodes. I have incorporated specific rules for binding of MS2 coat proteins, details of which can be found in the supporting information.

## Computational simulation of translational control in MS2

To demonstrate the types of translational control problems that can be examined with the ribosome kinetics model reported here, I examine a well studied problem of translational control in a viral mRNA, that of the translational coupling/repression mechanism in bacteriophage MS2. The purpose of this example study is to show that, by inputting only the sequence and initial secondary structure of bacteriophage MS2 mRNA into the model, the program is able to reproduce the translational control mechanism observed experimentally based on the folding response of the viral mRNA to ribosome movements. Thus the program is set-up to explore the protein expression and dynamics of a general mRNA, as long as a secondary structure has been either obtained experimentally or predicted computationally.

Fig 4A shows the start and stop codon locations relative to the secondary structure in a section of the bacteriophage MS2 mRNA, between nucleotides 1284 to 3569 in the wild-type

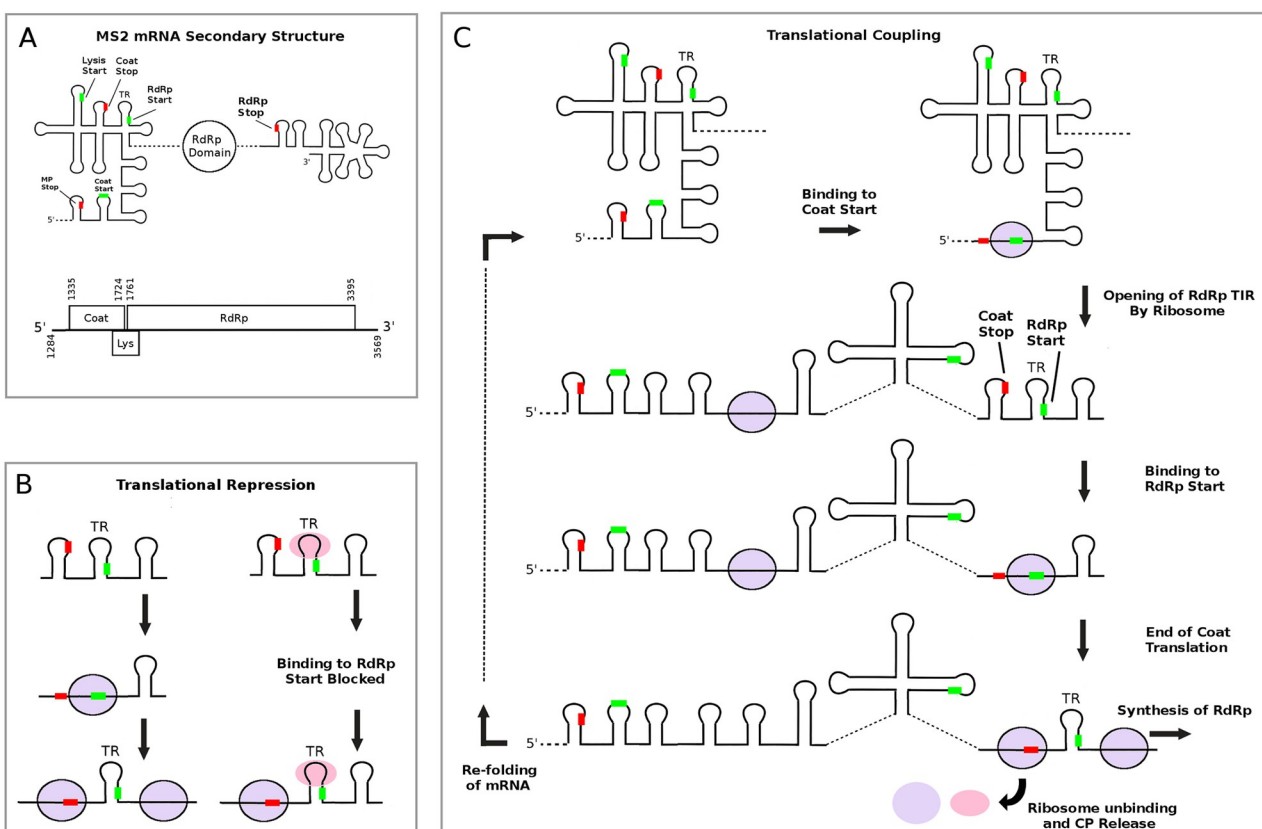

**Fig 4. The translational coupling and repression mechanisms in bacteriophage MS2 mRNA.** (A) Secondary structure cartoon of the bacteriophage MS2 coat and RdRp genes determined by phylogenetic analysis and enzymatic probing. (B) Translational repression of the RdRp gene occurs after synthesis of sufficient coat protein, which then binds to the translational repression (TR) hairpin that contains the start codon for the RdRp gene, blocking further ribosome initiations at this gene. (C) Diagram showing the translational coupling between the coat protein and RdRP genes. Synthesis of coat gene by the ribosome opens up a secondary TIR for the RdRp gene after melting of mRNA structure, allowing ribosome initiation at the RdRp start codon. Re-folding of the mRNA after translation of the coat gene hides the TIR for the RdRp gene.

sequence that encompasses the coat, lysis, and RdRp genes. The secondary structure of the entire MS2 mRNA has been determined using a combination of phylogenetic analysis combined with enzymatic probing [26, 27], and a cartoon diagram of the secondary structure is shown around the coat gene region. Fig 4B and Fig 4C illustrate the translational repression and coupling mechanisms of MS2, both of which have been determined from a variety of experimental studies over the past 30+ years [13, 24]. The translational coupling mechanism is illustrated in Fig 4C, where synthesis of RdRp is dependent (i.e. coupled) to expression of the upstream coat gene. The coat protein provides negative feedback to the expression of the RdRp gene, repressing ribosome binding to the TIR of the RdRp gene (c.f. Fig 4B).

Simulations of protein synthesis in the bacteriophage MS2 polycistronic mRNA were performed assuming a bacterial growth rate of $\mu = 0.7$ doublings per hour. At this growth rate, it is estimated that there are roughly 7500 active ribosomes in the *E. coli* cell, corresponding to a concentration of $12.5\mu M$ [28]. Specific concentrations of the various other proteins, tRNAs, etc. that are required by the ribosome can be computed from the ribosome concentration and the tables in the supporting information in [10] or this work. The predictions from the model are compared with experimental measurements done by Van Duin and colleagues on coat protein expression in bacteriophage MS2 [13]. It should be noted that these experiments were performed on a temperature sensitive plasmid where protein induction is triggered by raising the ambient temperature of the culture to 42° C. Thus, thermodynamic energies and unfolding rates for the RNA folding reactions where computed using Turner 99 energy parameters at $T = 42°C$ [21]. Currently, it is unknown how the increase in temperature would effect the kinetic rates for ribosome elongation. Since *E. coli* grows relatively similarly at 42°C as it does at 37°C where the kinetic parameters are taken from, I assume ribosome kinetic rates can be considered roughly unaltered by the temperature change. It would be potentially possible to scale the kinetic rates, following the method in Rudorf *et al.* [29]. However, this would require a measurement of one of the kinetic rates for ribosome kinetics at 42°C which is currently unavailable.

**Modelling the effects of mRNA mutations on coat protein synthesis.** Since the extended ribosome kinetic model takes into account the full nucleotide sequence of the MS2 mRNA along with its secondary structure, I can use it to examine how specific mutations to the mRNA result in subsequent changes in the levels of coat protein produced and derive kinetic parameters from experimental measurements of coat protein synthesis [13]. The predominate RNA structure which controls the overall coat protein synthesis rate is the 27 nucleotide coat hairpin (c.f. Fig 5A), which contains both the start codon and partial GGAG Shine-Dalgarno sequence. Van Duin and De Smit [13] have previously performed experimental measurements of the relative levels of coat protein produced for various coat hairpin mutants compared with the wild-type sequence. They found that the amount of coat protein produced was predominately determined by the free energy of the hairpin, and that destabilising the hairpin did not increase protein yields over that of the wild-type. This suggests that the coat gene is saturated, and that ribosomes binding to the TIR of the MS2 coat gene are not slowed by the time for the coat hairpin to melt. Assuming that the rate limiting step of protein expression is the rate of ribosome binding to the TIR (Eq 1), then a theoretical protein expression formula can be assumed to follow,

$$E = \frac{A}{1 + \tau_u B} = \frac{A}{1 + Ce^{-\beta \Delta G_F}} \, , \tag{2}$$

where $A$, $B$, and $C = B/k_F$ are constants to be determined. The constant A represents the maximal synthesis rate for the coat gene in the absence of any secondary structure in the TIR. The

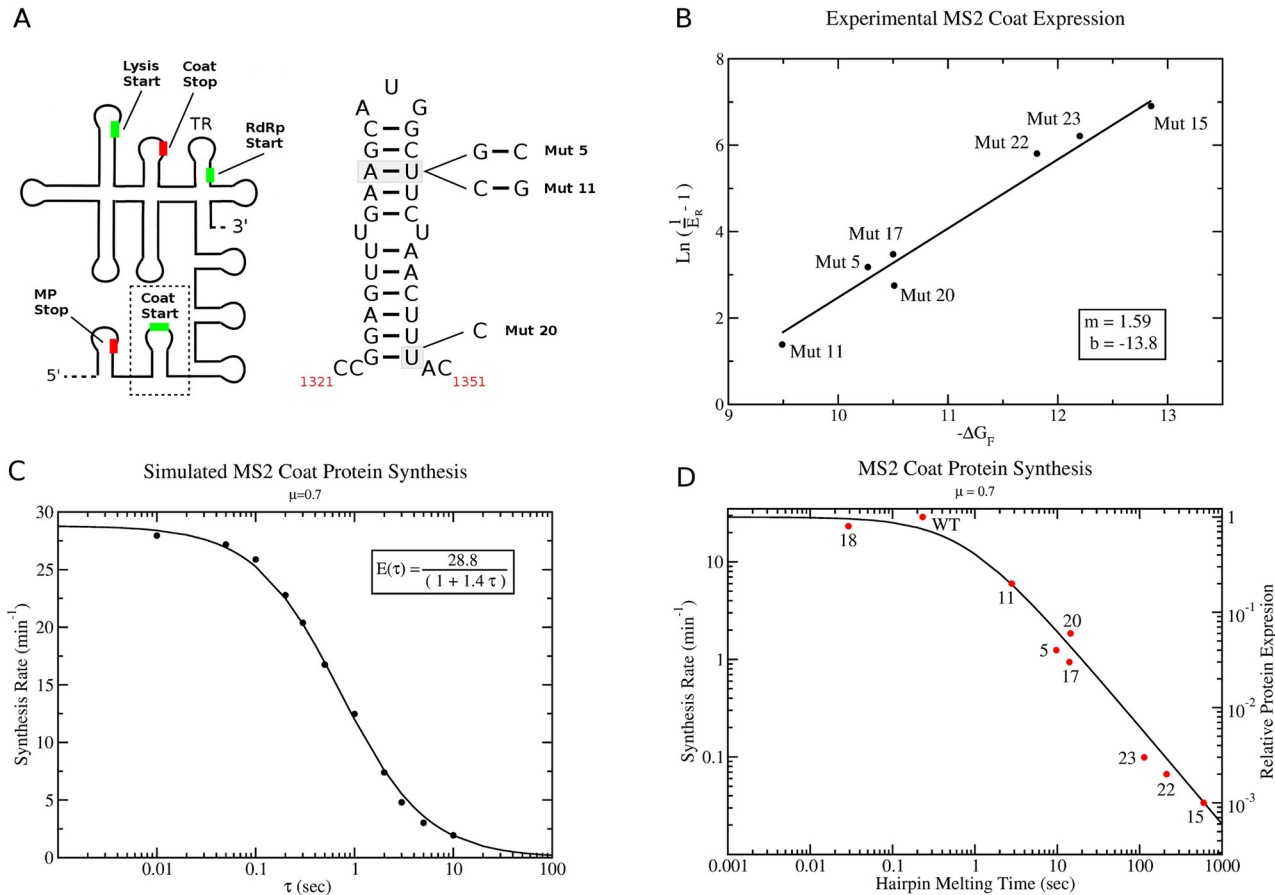

**Fig 5. Coat protein expression in bacteriophage MS2 for different hairpin mutants.** (A) Cartoon diagram of the secondary structure for the MS2 coat gene. Start codons for the various phage genes are indicated with a green bar, while stop codons are indicated with a red bar. The nucleotide sequence of the hairpin encompassing the coat gene start codon (dashed box in cartoon) is shown to the right. Mutants are labelled following [13]. (B) Best fit linear line of the experimental measurements from [13] to Eq 4. (C) Predicted MS2 coat protein synthesis rates are calculated from 30 min of ribosome kinetics at a growth rate of $\mu = 0.7$ doublings per hour using hairpin unfolding times ranging from $\tau_u = 0.01$ to $\tau_u = 10$ seconds (black dots). The best fit of the data to the theoretical protein expression curve (Eq 2) is represented by the black line. (D) Fit of experimental relative coat protein expression data (red dots) to the theoretical expression curve (Eq 2).

factor B represents the kinetic rate of hairpin unfolding when protein expression is half of the maximum value, while C is obtained from B via the relation

$$\tau_u = \frac{e^{-\beta \Delta G_F}}{k_F} ,\tag{3}$$

which estimates the hairpin unfolding time from the kinetic rate of hairpin folding, $k_F$, and the thermodynamic free energy of the hairpin $\Delta G_F$. It is important to note that both the A and B coefficients in Eq 2 will be dependant on the full translational dynamics of the cell, i.e. concentration of ribosome subunits, ternary complex, growth rate, codon bias of the mRNA (as noted in [16]) etc. Although there are some experimental measurements of hairpin folding kinetics [30], in general there is limited information for estimating values of $k_F$ for general RNA hairpins. However, the experimental data from Van Duin and De Smit on the relative expression of coat protein for a variety of hairpin variants [13] allows the fitting of the $k_F$ parameter using my model. Reformulating Eq 2 in terms of the coat protein expression relative to the

**Table 1. Predicted and measured relative coat protein expression in bacteriophage MS2 for different coat hairpin mutants.** The relative coat protein expression ($E_R$) of hairpin mutants are shown relative to the peak protein expression. Measured values of $E_R$ are obtained from [13], while theoretical values are from Eq 2. The hairpin unfolding times $\tau_u = e^{-\beta\Delta G}/k_F$ are estimated using a hairpin folding rate of $k_F = 1.36 \times 10^6 s^{-1}$ and $\beta = 1.59$ mol/kcal. Hairpin $\Delta G_F$ values are calculated using Turner 99 energy parameters at $T = 315K$ (42°C) [21].

| Mutant | $\Delta G_F$ | $\tau_u$ | E (min$^{-1}$) | $E_R$ (Measured) | $E_R$ (Theory) |
|--------|--------------|----------|----------------|------------------|----------------|
| 18 | -6.63 | 0.0291 | 27.672 | 0.800 | 0.960 |
| wt | -7.93 | 0.2319 | 21.739 | 1.000 | 0.755 |
| 11 | -9.49 | 2.8005 | 5.853 | 0.200 | 0.203 |
| 5 | -10.27 | 9.7305 | 1.969 | 0.040 | 0.068 |
| 17 | -10.50 | 14.0485 | 1.393 | 0.030 | 0.048 |
| 20 | -10.52 | 14.5044 | 1.351 | 0.060 | 0.046 |
| 22 | -11.81 | 113.781 | 0.179 | 0.003 | 0.006 |
| 23 | -12.20 | 212.090 | 0.097 | 0.002 | 0.003 |
| 15 | -12.85 | 598.787 | 0.034 | 0.001 | 0.001 |

maximum i.e. $E_R = E/A$ and taking the natural logarithm one obtains

$$\ln\left(\frac{1}{E_R} - 1\right) = \ln\left(\frac{B}{k_F}\right) - \beta\Delta G_F. \tag{4}$$

Fig 5B shows the experimental data from Van Duin and De Smit [13] for the hairpin variants with <100% relative coat protein expression. Assuming $m = \beta = 1/k_b T = 1.59$ mol/kcal for $T = 315$ K (42°C), the best fit line to the data ($y = mx + b$) gives an intercept of $b = \ln(B/k_F) = -13.8$. Simulating a variety of hairpin melting times through explicit variation of $\tau_u$ in the stochastic ribosome model, I obtain the expression curve shown in Fig 5C, which can be fitted to Eq 2 to give best fit values of $A = 28.8$ min$^{-1}$ and $B = 1.40$ sec$^{-1}$. This allows me to use the experimental data to obtain an estimate for $k_F$ for the coat hairpin of $k_F = 1.36 \times 10^6$ sec$^{-1}$. Using this value of $k_F$, one can obtain the expected unfolding times $\tau_u$ for the various hairpin mutants and the wild-type coat hairpin (Table 1).

Table 1 summarises the experimental data from [13] for mutants which have relative expression less then 1. The theoretical expression curve obtained from the stochastic simulations in Fig 5C is overlaid with the experimental data from Van duin (red dots) in Fig 5D as a log/log plot with maximum theoretical coat protein synthesis rate of 28.8 proteins per minute. As can be seen, the theoretical curve gives a good fit to the experimental data. One possible explanation for the differences observed is that at high concentrations of coat protein, it may be difficult to accurately measure relative coat protein concentration using a western blot as was done in [13]. Since error bars were not reported in [13], the extent to which the theoretical expression curve falls within the experimental variance is unknown. However as a further check, one can compare the estimates for hairpin unfolding times in Table 1, which are implied by the experimental data, with the theoretically predicted hairpin unfolding times for the wild-type hairpin using the RNA kinetics folding program KFOLD [31]. The KFOLD calculation of the mean first passage time for unfolding ($\tau_u = 0.18$ sec) is in close agreement with the estimate from the experimental fit in Table 1 ($\tau_u = 0.23$ sec).

**Modelling translational repression and coupling.** In addition to examining how nucleotide mutations to the mRNA can alter the stability of the TIR and subsequent protein synthesis rates, my model can also examine the phenomenon of translational coupling and translational repression. The translational coupling observed in bacteriophage MS2 is between the coat and RdRp genes, illustrated in Fig 4C, where synthesis of the RdRp gene is dependant on synthesis of the coat protein. As levels of coat protein accumulate in the cell, coat protein binds to the

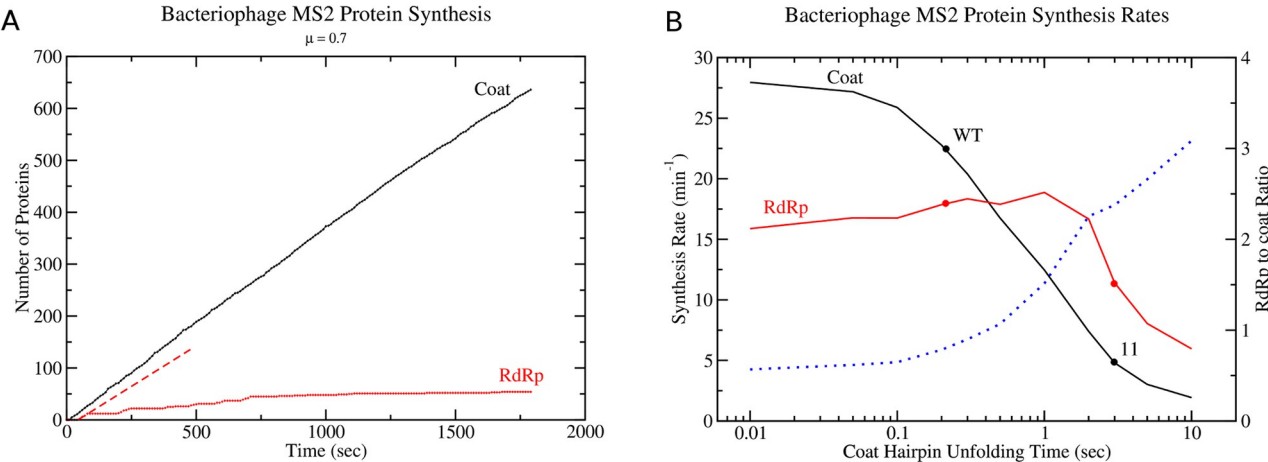

**Fig 6. Synthesis rates for Coat and RdRp proteins in Bacteriophage MS2.** (A) Temporal dynamics of coat protein and RdRp synthesis from bacteriophage MS2 mRNA in a bacterial cell over 30 minutes. Red dashed line corresponds to the initial synthesis rate of RdRp before coat protein binding to the TR stem-loop (c.f. Fig 4A) suppresses further synthesis. (B) Maximal Coat and RdRp synthesis rates from bacteriophage MS2 mRNA as a function of coat hairpin unfolding time. The curve is computed by varying $\tau_u$ explicitly in the program, with dots representing the results from simulations with specific mutant sequences. Blue dashed line shows the ratio of RdRp to Coat protein synthesis rates.

TR stem-loop (c.f. Fig 4A and Fig 4B), repressing translation of the RdRp gene by ribosomes. Using the detailed rules for coat protein binding kinetics to TR hairpin (see supplementary information) incorporated into the model, I have simulated 30 min of protein synthesis for the RdRp and coat proteins in a cell growing at $\mu = 0.7$ doublings per hour. Fig 6A shows the resulting numbers of coat and RdRp proteins in the cell.

The maximal synthesis rate of RdRp closely follows that of coat protein, before binding of coat protein to the TR stem loop suppresses further synthesis. Fig L in S1 Text illustrates a snapshot of the ribosome density and mRNA secondary structure both before translational repression of RdRp by coat protein and after. As can be seen in Fig L, the mRNA structure of the RdRp domain re-folds after translational repression and only the coat gene is saturated with translating ribosomes. Since the RdRp gene is coupled to the coat protein, one can examine how RdRp synthesis is affected by ribosome initiations on the coat gene (or equivalently the coat hairpin melting time). Interestingly, a plot of the maximum RdRp synthesis rate versus the coat hairpin unfolding time, $\tau_u$, reveals that RdRp synthesis is maximised for coat hairpins with unfolding times ranging between 0.2 and 2 seconds (c.f. Fig 6B). Moreover, as coat hairpin unfolding times increase, the RdRp/Coat synthesis ratio (blue dashed line Fig 6B) increases to a maximum of around 3. Since coat and RdRp TIRs are separated by approximately 420 nucleotides, this ratio corresponds to the number of ribosomes that are able to load onto the RdRp start codon (approximately 3–4) within the time it takes for a ribosome to finish translating the coat gene and the RdRp TIR is once again sequestered into RNA secondary structure. This demonstrates the importance of taking into account the melting of RNA structure and exposure of downstream TIRs, as well as translation time of the ribosome on the upstream gene when predicting protein synthesis rates for translationally coupled genes.

## Discussion

In this work, I have demonstrated a new stochastic simulation tool for modelling ribosome kinetics on dynamic mRNAs. This computational tool allows for the effects of mRNA folding

and interactions of the mRNA with proteins to be taken into account, allowing temporal protein expression profiles and feedback mechanisms at the translational level to be studied theoretically. Application of the model to the example system of bacteriophage MS2, a (+)ssRNA virus which infects *E. Coli*, shows that the model is able to recapitulate the coat protein synthesis ratios of different coat hairpin mutants which were measured experimentally by Van Duin and De Smit [13]. Although Van Duin and De Smit have previously described a thermodynamic equilibrium model of coat protein expression based on hairpin $\Delta G_F$ values, my work here extends this work to obtain theoretical estimates for the kinetic parameters for ribosome binding to the TIR, along with kinetic parameters for hairpin folding, while creating a general model for examination of translational coupling and repression in mRNAs. I have demonstrated that my kinetic model for 30S binding to the TIR is able to predict the relative coat protein expression, which depends on the relative stability of the local RNA fold of the coat-hairpin, for a variety of hairpin mutants that is consistent with Ref. [13].

In addition, my model has been able to make several predictions for the behaviour of RdRp synthesis (c.f. Fig 6) as the thermodynamic stability of the coat hairpin ($\Delta G_F$) is altered. Interestingly, it suggests that weaker coat hairpins reduce RdRp synthesis rates, due to ribosomal queuing at the coat gene stop codon which blocks ribosome access to the RdRp TIR. This effect results in a gradual ≈15% decrease in the synthesis rates for RdRp as coat protein synthesis becomes a maximum for weaker coat hairpins corresponding to shorter unfolding times (c.f. left hand side of Fig 6B). Previous models of translational coupling [32] predict that high expression of up-stream genes are expected to result in a plateau in the expression of downstream translationally coupled genes, which is in contrast to this observation here. While Tian and Salis did note that ribosomal queuing was a potential issue for blocking ribosome initiations on downstream genes, my model suggests that the effect is more pronounced then expected. Moreover, this reduced expression of RdRp for weaker hairpins may explain why mutational studies of MS2 coat hairpin [33] show that both weaker and stronger coat hairpins than the wild-type coat hairpin were selected against. One explanation from the model is that coat hairpins with wild-type stability optimise both coat and RdRp synthesis and prevent queuing ribosomes from blocking access to the RdRp TIR. This is consistent with evolutionary experiments which show weaker coat hairpins result in decreased viral plaque formation [33]. These observations made by the model of RdRp expression depend on the global fold and overall expression of the coat gene, as the expression of RdRp is translationally coupled to the coat gene. Thus, new experiments will be required to validate these results and the overall RNA co-translational folding predictions made by the model.

However, while Tian and Salis developed a translational coupling model which accounts for both *de novo* ribosome initiation and re-initiation, at the moment my model only accounts for *de novo* ribosome binding. Green and colleagues [34] have noted that translational coupling is still observed in *E. Coli* mutants which are unable to express recycling factor, suggesting that *de novo* ribosome binding is the main source of translation initiation in coupled genes. It should be noted however, that translational re-initiation is still a phenomenon that requires proper consideration. For example, it is believed that ribosome initiation at the bacteriophage MS2 lysis gene is triggered by ribosomal scanning and re-initiation after synthesis of the coat gene [35]. Despite these shortcomings, the computational techniques developed here to model the dynamic response to mRNA structure and protein synthesis levels as the ribosome melts mRNA structures can provide the synthetic biology and ribosome communities an important tool for examining translational coupling and translational repression in both synthetic and natural mRNAs. In particular, it may be a useful tool for examining the regulation of (+)ssRNA viral infection dynamics in a cell.

## Methods

### Model parameterisation

The kinetic rates used in the model are identical to those from my previous ribosome model [10], with the exception of the initiation rates for 30S PIC binding and standby site formation. Specific adjustments are discussed in the supporting information. Similarly, the protein abundances for Ef-Tu, Ef-Ts, etc. are identical to those found in [10], with the exception of the tRNAs, which were optimised to match codon bias present in *E. coli* K12 strain MG1655 (see supporting information for details of the optimisation procedure). In addition, RF1 and RF2 concentrations were also slightly adjusted to reproduce experimentally observed tRNA misreading rates [36]. The transcriptome for the background of cellular mRNAs constructed using authentic mRNAs from *E. coli* K12 strain MG1655 following the procedure listed in supporting information. Software used in the simulations can be downloaded from http://www-users.york.ac.uk/∼ecd502/ or from Github at edykeman/ribofold.

## Supporting information

**S1 Text. Supplementary information. Fig A: Tree representation of the secondary structure of the bacteriophage MS2 coat gene.** Secondary structural elements (hairpins and multi-loop helices) are labelled 1–15, with 0 used to label the exterior loop. The tree representation of the secondary structure is shown in the upper left, with each node of the tree representing one of the helix elements, i.e. a hairpin or multi-loop helix. Black arrows indicate a linked list pointing to the 5' and 3' neighbours of each structural element. Green arrows point to leaf nodes, the 5' most helix element in a multi-loop, while red arrows point to root nodes, i.e. the multi-loop helix which closes the multi-loop that the node is apart of. **Fig B: Tree representation of the secondary structure of the bacteriophage MS2 coat gene with bound ribosome.** Secondary structural elements (hairpins and multi-loop helices) and ribosomes bound to the mRNA are labelled 1–15, with 0 used to label the exterior loop. Yellow nucleotides indicate the footprint of the 70S ribosome while purple nucleotides indicate the location of the ribosome P-site. The tree representation of the secondary structure is shown in the upper left, with each node of the tree representing either a helix element or bound ribosome. Links between nodes (black,green and red arrows) follow the same rules as in Fig A. **Fig C: Example of how a local hairpin RNA folding transition is constructed.** The bacteriophage MS2 coat gene and its tree representation are given with each node in the tree representing either a helix or ribosome. The yellow shaded nucleotides give the ribosome footprint on the mRNA while the purple nucleotides denote the location of the P-site. The blue nucleotides colour the nucleotides which make up the window fragment. This window is extracted and the lowest energy RNA fold computed. The window fragment is replaced with the lowest energy fold to construct the new RNA fold, and the folding transition rate ($k_F$) is computed using a breadth-first-search barrier prediction algorithm. **Fig D: Construction of folding windows.** (a) Example of how folding windows are constructed for a simple RNA structure consisting of three hairpins and no multi-loops. Here $N_w = 2$ and two folding windows are constructed for each node in the tree. Red and green arrows depict how sections of the RNA are extracted for each window (example for node 0 shown). (b) Example of folding window construction on the same RNA structure as in (a), but with $N_w = 4$. For this setting and RNA fold, all possible window fragments will be considered. **Fig E: Construction of folding windows with $N_w = 4$ for an RNA containing multi-loops.** (a) Example of how folding windows are constructed for the exterior loop (region 0). Folding windows are not allowed to contain a multi-loop, ribosome, or protein bound RNA structure, hence folding window 4 is empty for node 0. (b) Example how folding

windows are constructed for the multi-loop (region 3). Folding windows follow the same construction procedure as for the exterior loop. **Fig F: An RNA Transition Pathway Between Two Secondary Structures** The secondary structural states that the RNA transitions through are denoted by $\mathcal{S}_i$ and the kinetic rates for moving forward or backward along the path are given by $k_i^+$ and $k_i^-$, respectively. **Fig G: Pseudo-code for various RNA path finding algorithms** (a) *Greedypath*. Pseudo-code for the prediction of the optimal RNA transition path using the greedy method of Voss [37]. (b) *Findpath*. Pseudo-code for the prediction of the optimal RNA transition path using the breadth-first search method of Flamm [19]. (c) *Findpath-mfp*. Pseudo-code for the prediction of the optimal RNA transition path using the breadth-first search method of Flamm [19], but with paths selected according to those having the lowest mean first passage times. **Fig H: Kinetic model of 30S:PIC binding to mRNA to form the 30S Initiation Complex.** The initial binding of 30S:PIC to mRNA with rate $k_1^B$ proceeds via recognition of ribosomal protein S1 (yellow dot) followed by recognition of the Shine-Dalgarno sequence and start codon. The model includes two pathways to formation of the 30S:IC: (1) the standby pathway, where the 30S:PIC first binds to a weakly structured area of the mRNA and waits until RNA unfolding presents the start codon, and (2) a pathway in which the 30S:PIC binds to unstructured RNA. **Fig I: Energy profile of 30S:PIC binding to mRNA.** (A) The binding of the 30S:PIC subunit to mRNA can take place via two pathways, one which depends on structured mRNA where 30S:PIC binds via a standby site (State 2) and a second, where mRNA is mostly unstructured and the 30S:PIC skips a standby state. (B) Model of the energy profile of 30S:PIC interaction with the mRNA. In states 2 and 3, the 30S subunit is modelled as interacting with mRNA predominately via ribosome protein S1 (yellow dot), where in state 4, the interaction is directly with the start codon (green bar = start, red bar = stop) and any Shine-Dalgarno sequence present in the mRNA. **Fig J: Apparent kinetic rates of 30S:PIC binding to mRNA.** (A) Plot of Equation 16 for varying TIR melting times. The free concentration of 30S:PIC was set to 0.69 $\mu$M, following the predicted estimates from my Ribosome model [10] at a growth rate of $\mu = 0.7$ doublings per hour. (B) Plot of Equation 17 for varying strength of the interaction between the Shine-Dalgarno sequence with rRNA ($\Delta G_4$) using values $k_{-1}^B = 10$ s$^{-1}$ and $\Delta G_{S1} = -9.19$ kcal/mol with $\Delta x = \Delta G_4 - \Delta G_{S1}$. **Fig K: The three families of RNA hairpins which bind bacteriophage MS2 coat protein.** (A) The TR, F6, and F7 hairpin variants which have been shown to bind MS2 coat protein. Sequence preferences are denoted in the figure with N = any nucleotide, Y = Pyrimidine, R = Purine. (B) Affinity matrix estimated from stop flow kinetic assays on binding of MS2 coat protein to the TR hairpin [38]. The matrix can be used to estimate the change in binding affinity that results from sequence or structural changes. **Fig L: Snapshots of MS2 mRNA secondary structure of the coat and RdRp genes during translation.** (A) Secondary structure of MS2 mRNA at low coat protein concentrations when both the coat and RdRp genes are being actively translated by ribosomes. Yellow nucleotides indicate the footprint of ribosomes while purple nucleotides indicate the location of the ribosome P-site. Important secondary structures (coat hairpin, TR stem-loop, RdRp Stop Hairpin) are labelled. Numbers indicate nucleotide number in the MS2 viral RNA. (B) Secondary structure of MS2 mRNA at high coat protein concentrations when only the coat gene is being actively translated by ribosomes. The structure of the RdRp gene has re-formed long-distance interactions and has no actively translating ribosomes, while the coat gene has three ribosomes in active translation. **Table A: Numbers of various tRNAs per cell and their codon recognition.** Data for the number of tRNAs at different growth rates have been adjusted to match the codon biases of the mRNAs from E. coli K12 (strain MG1655 —uniprot accession code U00096). The total tRNA at each growth rate have been normalised to overall expected total tRNA concentrations discussed in Bremer [28]. **Table B: Predicted**

**and measured tRNA$^{Lys}$ misreading frequencies.** The misreading frequency by tRNA$^{Lys}$ per 10000 reads at various near-cognate codons is given. Experimental measurements are obtained from [36] and are compared with the model at different growth rates. **Table C: Mean first passage times for MS2 coat hairpin unfolding.** The mean first passage times ($\tau$) are calculated using two different methods (1) a calculation using 5000 simulations of KFOLD, (column KFOLD in table) and (2) a calculation using the breadth-first search algorithm in Fig G(c) (column BFS in table). Temperatures are in degrees Celsius while mean first passage times are in seconds.
(PDF)

## Acknowledgments

ECD wishes to thank Prof. R. Twarock and Dr. R. J. Bingham for careful reading of the manuscript and their helpful feedback.

## Author Contributions

**Conceptualization:** Eric C. Dykeman.

**Data curation:** Eric C. Dykeman.

**Formal analysis:** Eric C. Dykeman.

**Funding acquisition:** Eric C. Dykeman.

**Investigation:** Eric C. Dykeman.

**Methodology:** Eric C. Dykeman.

**Project administration:** Eric C. Dykeman.

**Resources:** Eric C. Dykeman.

**Software:** Eric C. Dykeman.

**Validation:** Eric C. Dykeman.

**Visualization:** Eric C. Dykeman.

**Writing – original draft:** Eric C. Dykeman.

**Writing – review & editing:** Eric C. Dykeman.

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
