## [Decision Letter · Decision Letter 0]

25 Nov 2022

Dear Dr. Dykeman,

Thank you very much for submitting your manuscript "Modelling ribosome kinetics and translational control on dynamic mRNA" for consideration at PLOS Computational Biology.

As with all papers reviewed by the journal, your manuscript was reviewed by members of the editorial board and by several independent reviewers. In light of the reviews (below this email), we would like to invite the resubmission of a significantly-revised version that takes into account the reviewers' comments.

In addition to the issues raised by the reviewers concerning clarity of presentation and model validation, please also pay attention to the problems regarding typographical errors in the text. 

We cannot make any decision about publication until we have seen the revised manuscript and your response to the reviewers' comments. Your revised manuscript is also likely to be sent to reviewers for further evaluation.

Sincerely,

Martin Meier-Schellersheim

Academic Editor

PLOS Computational Biology

Mark Alber

Section Editor

PLOS Computational Biology

Reviewer's Responses to Questions

**Comments to the Authors:**

Reviewer #1: In this manuscript Eric Dykeman develops a computational model of co-translational folding of messenger RNAs and compares the predictions of the model to experimental data on translation of two translationally coupled proteins of the MS2 virus. The model appears novel and its results make it clear that the interplay of RNA structure kinetics, ribosome initiation, and ribosome motion can lead to unexpected consequences and thus has to be comprehensively modeled to obtain a quantitative understanding of these processes. Thus, this model is in principle valuable. However, the manuscript as written has many shortcomings, is difficult to follow, and leaves important questions open. It thus requires significant revisions before it can be reevaluated.

Major issues:

1) There are many many bells and whistles in this model. On the other hand, there seems to be fairly little actual comparison to experimental data in this manuscript (see also point 2) below). It would thus be important to know, which of these details actually matter and which would allow other choices (or omission altogether) without degrading the comparison to experimental data. These are:

1a) The model considers ribosomes not only on the one structured RNA of interest but also on all other mRNAs in the cell (albeit treating them as without structure). Is this really necessary? It seems that this results in spending very significant computational resources on what probably mostly amounts to renormalized concentrations of available ribosomes and tRNAs.

1b) Why are overlapping TIRs pruned to just the one with the lowest interaction free energy? Aren't all TIRs in principle available to the 30S subunit to bind to?

1c) Is actually anything beyond the hairpin containing the start codon relevant (see also 2b) )

1d) There seem to be a lot of parameters going into the description of 30S binding. Are the results sensitive to all of these?

2) Validation of the model by comparing it with experimental data is of course key for any modeling paper. However, there are various issues with this validation (see also point 9) ):

2a) What is actually compared to experiment? It seems that the general form of equation (2) is independent of all the model details. So, is the only thing that actually comes out of running this entire highly complex model the values of the parameters A and B? This seems to be very little information to claim validation based on given all the many parameters that go into the model.

2b) Do lines 340-347 really say that the only thing that really matters for this comparison with the experiment is the unfolding time of the coat hairpin and that KFOLD actually gets this parameter mostly right? If that is true, it does not seem as if this comparison to experiment really provides a validation of the model as a whole.

2c) Figure 5B is actually not very useful for assessing the quality of the agreement between the experimental data (red dots) and the theoretical line. The two data points at high relative protein expression (which the author tells the reader to ignore in the comparison) "squeeze" the actually important data points into a narrow range of y values that makes deviations appear very small. Indeed, e.g., mutant 22 is off by a factor of 2 according to table 1, which is of course not visible in figure 5B since both, the measured and the theoretical value are small. Plotting the measured and theoretical values from table 1 against each other, preferably on logarithmic axes would provide a more honest comparison.

3) The paragraph in lines 239-252 says that it is possible to parametrize the model such that the range of off rates agrees with the range of experimentally measured off rates. However, the ranges themselves cover many orders of magnitude. Is there agreement with the experimentally measured off rates on a sequence by sequence level (which would be a much stronger statement) or is the agreement really only the general range?

4) The author's argument for why a coarse grained approach of the RNA kinetics is appropriate is that the time scale of ribosome movement is a slow 50ms, which is thus slower than (most) RNA folding kinetics. While this is true, another important aspect pointed out in the abstract is that the author includes protein RNA interactions. Wouldn't those be on faster time scales and thus might not be well described by this coarse grained approach?

5) In line 73 the author says that the model treats the RNA folding "at the single nucleotide level". Yet, later it becomes clear that the RNA folding is treated in a coarse grained way. The author should weaken the claim in the introduction accordingly, so as to not confuse the casual reader.

6) In line 272 it is stated that the input to the model is the sequence and the structure of the RNA. It is somewhat surprising that the structure is necessary and it is not very clear in how far this provided structure constrains the potential structures the RNA visits during the simulations. Would alternative structures that are not subsets of the provided structure be forbidden? Probably not, since the methods do talk about refolding subsequences of the RNA. But the set of alternative structures is somehow limited by the input structure, isn't it? The manuscript should spell out crystal clear in which way the set of possible structures visited during the simulation is limited by the input structure. This seems to be a key point of the whole method (and a possible weakness in cases of RNAs with multiple alternative structures).

7) The finding of translational coupling between the coat protein and RdRp is quite interesting. However, it seems that the comparison to experiment here is purely qualitative. Is there a way to make more quantitative statements? After all, the abstract says that the model "accurately reproduces" this behavior. What does "accurately" mean if no numbers are compared between experiment and modeling? Can it at least be made clear which of the findings are predictions for (potential) future experiments and which are actually compared (favorably) to existing experiments? This entire section is also somewhat difficult to understand. It should spell out explicitly what type of protein binding is included. Is every hairpin a potential binding site for a coat protein? Or just one? And how was figure 6B obtained (specifically what was actually varied to obtain the different hairpin unfolding times? Some model parameter? The hairpin sequence?)

8) Could the author comment on the discrepancies observed in table S3, especially the one for mutant 11 at 37 oC? Does this factor of 3 matter?

Minor issues:

9) Something went really wrong with the paragraph presenting the comparison with the experiment (below equation (4)). All the figure references here are to figure 3. At least most of them should probably be to Figure 5 instead? But maybe not the first one? And why are the panels in figure 5 in the order in which they are? Shouldn't D precede B logically? That also seems the order in which they appear in the text. This is really the key section of the manuscript since it is the comparison to the experimental data that is supposed to validate the model and confusing the reader here by sloppy figure references is really just a bad idea.

10) The discussion around equation (1) is very difficult to follow, especially without first reading the supplemental material. Please provide enough context that the logic behind what is presented here becomes clear without knowing all the methodological details already.

11) Another difficult to read section is the discussion of the translational results in lines 394-402. Talking about "plateauing" is counter-intuitive if said "plateauing" is only seen when reading figure 6B from right to left and it is also left to the reader to make the connection that "weaker" hairpin means shorter unfolding times (which is the x axis on figure 6B).

12) It is a strange choice to leave essentially the entire description of the algorithm to the supplement and have just one paragraph of methods in the main text. After all, the whole manuscript is primarily about the model. And there is no length restriction in this journal, is there? The author should consider moving some of the supplemental material from the supplement into the main text. While doing so, it would also be helpful to be more explicit about what the allowed steps in the Gillespie algorithm are, which is somewhat implicit in this current version.

13) The author should try to be cognizant of significant digits. Is the temperature quoted in line 329 to five significant digits really controlled to that precision in the experiments that are referred to? Is the intercept in line 330 really reliable to 5 significant digits?

14) The choice of using the capital letter K for rates seems non-standard. Typically, capital K stands for equilibrium constants while rates are denoted by lower case k.

15) There are a lot of typographical errors in the manuscript, which make the entire work appear very sloppy. It should be standard practice to at least spell check a manuscript before submission:

- abstract: "co-translational fold" -> "co-translationally fold"

- line 23: "dependant" -> "dependent"

- line 36: "their" -> "there"

- line 37: "uses" -> "use"

- line 38: "its" -> "their"

- line 148: "to computed" -> "to be computed"

- line 210: "week" -> "weak"

- line 214: "specifcially" -> "specifically"

- line 240: "then" -> "than"

- line 268: "imputing" -> "inputting"

- line 298: "42 degrees C" -> turn "degrees" into degrees symbol

- line 299: "42 C" -> add degrees symbol

- line 303: "results" -> "result"

- line 307: "Shine-Delgarmo" -> "Shine-Dalgarno"

- line 362: "snapshpot" -> "snapshot"

- line 367: "effected" -> "affected"

- line 399: "platues" -> "plateaus"

- line 401: "platue" -> "plateau"

- Table 1: add index "F" to Delta G

- Supplemental material page 1: "M61655" -> "MG1655"

- Supplemental material page 3: "structure needs accounts" -> "structure accounts"

- Supplemental material page 12: "16s" -> "16S"

- Supplemental material page 12: "M" -> "mol" (molar and mole are different!)

Reviewer #2: Review of PCOMPBIOL-D-22-01546

Dr. Dykeman developed an algorithm for the simulation of protein synthesis rates in E. coli. His approach is unique in the sense that he combines a very detailed model of mRNA translation with mRNA folding dynamics to assess translation initiation rates.

Overall, the manuscript is well written and has a clear structure. The results are well presented and the reader can understand how they were obtained. The presented work is a very nice achievement and its applicability might only be hampered by the lack of experimental data on in-vivo mRNA secondary structure (which should not be of relevance for assessing a computational model). I recommend publication after the following concerns have been addressed.

Major concerns:

1) The author needs to put his work in a broader context of what has been achieved before. There is a thorough discussion only of refs. 11 and 26 (if I did not miss anything). But a quick internet search reveals for example the paper PMID: 20504310 that also claims to introduce a “model [that] uses mRNA-folding dynamics and ribosome-binding dynamics to estimate translational efficiencies solely from mRNA sequence information”. The author should inform the reader of more approaches alternative to his and give a hint how these approaches differ from the presented one.

2) The same concern applies to mathematical models and simulations of mRNA translation in general. I understood that the author here presents a further development of his previous algorithm. Nevertheless, it is necessary (and informative for the reader) that he briefly compares (or advertises) his underlying model for computation of translation rates to that of other researchers. A detailed mechanistic model has been applied for example in PMID: 31101858 and PMID: 31369559, and there are a few more examples in the literature of detailed models and more simple measures (like tAI or CAI).

3) Further citation is needed, for example on lines 31/32 (statement), 36-39 (statement), 290/291 (parameter values), caption of Table 1 (to Turner 99). The author should check if he missed further citations.

4) Code availability: Currently, the software is available from the author’s homepage and the FORTRAN code is accessible after download. For a better general accessibility, findability, reusability and visibility (among others) it would be favorable to host the code on a public repository (such as Gitlab, Github). This would also enable direct interaction with users via creation of issues or even further development of the code by the community. (The author might even think about developing a SBML model to be published at the BioModels database if that is possible.) See also journal’s statement: “The data and code should be provided as part of the manuscript or its supporting information, or deposited to a _public_ repository”.

I have a few more questions and would be glad if the author could comment and/or adjust the text accordingly:

1) Are the predicted TIRs confirmed by known start-codon positions?

2) Line 201: Why these codons (as alternative start codons) but not AUU?

3) Line 206: Why are TIRs removed from the list based on \\delta G, i.e., “binding strength” of the 30S:PIC? Is there experimental evidence that TIRs are optimized in that sense?

4) Line 299: Where rates also corrected for temperature or are the differences for T = 42°C vs. 37°C considered to be too minor?

5) Eq. 2): Could you please elaborate? What are the meanings (not just definitions) of the constants A, B, C?

Further minor issues:

1) The manuscript could be written more concisely. See for example repetitions in line 268 (p. 7) vs. line 272 (p. 8) or line 282 (p. 8) vs. 284 (p. 8).

2) The manuscript needs thorough typo correction, for example lines 327-338 refer to Fig. 3 instead of Fig. 5.

3) Line 193 “appearing and disappearing”: Is the TIR “disappearing” or just nor accessible?

4) Lines 249 and 251: The ranges [0.001, 10.0] and [0.001, 4] differ; in which sense is this “consistent”?

5) Line 139: The term “co-translational folding” is used differently (depending on context; referring to folding of the nascent chain). To avoid confusion, I suggest to specify the term, e.g., “co-translational mRNA folding”

6) Line 326: I would use “maximum” instead of “peak”.

7) Line 337, “excellent fit”: Did ref. 11 report error bars (especially concerning #18 and WT)?

**Have the authors made all data and (if applicable) computational code underlying the findings in their manuscript fully available?**

Reviewer #1: Yes

Reviewer #2: Yes

PLOS authors have the option to publish the peer review history of their article (what does this mean?). If published, this will include your full peer review and any attached files.

Reviewer #1: No

Reviewer #2: No
---

## [Decision Letter · Decision Letter 1]

31 Dec 2022

Dear Dr. Dykeman,

Thank you very much for submitting your manuscript "Modelling ribosome kinetics and translational control on dynamic mRNA" for consideration at PLOS Computational Biology. As with all papers reviewed by the journal, your manuscript was reviewed by members of the editorial board and, in this case, by two independent reviewers. The reviewers appreciated the attention to an important topic. Based on the reviews, we are likely to accept this manuscript for publication, providing that you modify the manuscript according to the review recommendations.

Sincerely,

Martin Meier-Schellersheim

Academic Editor

PLOS Computational Biology

Mark Alber

Section Editor

PLOS Computational Biology

Reviewer's Responses to Questions

**Comments to the Authors:**

Reviewer #1: The author has taken into account most of the reviewers' criticisms in this revision and the manuscript has thus improved. It tells the story quite well now, but there are still a few remaining points:

1) The author provides an explanation of why it makes sense to model translation on all mRNAs rather than only on the mRNA of interest in the reply to the reviewers. Please add some of this useful explanation to the actual manuscript as well so that readers can benefit from it.

2) This reviewer is still not convinced by the author's answer to the original point 2b). The question was about what the comparison to the experiment is really probing. The manuscript showed that running KFOLD on the TIR hairpin essentially gives the correct results. Yet, the author's method looks at the folding of the entire mRNA. This reviewer's concern was that the fact that KFOLD gives more or less the right answer implies that *none* of the folding of the rest of the mRNA matters for the comparison to the experiment. Thus, the quantitative agreement with the experiment does not really test anything about the author's algorithm other than the implementation of the folding of the TIR hairpin. If that is so, the author must acknowledge that in the discussion.

3) The explanation of the comparison to the Joseph and Studer results is also not satisfactory. Is the author saying that Joseph and Studer did not really observe a range, but rather did two experiments, one with a Shine-Dalgarno sequence and one without a Shine-Dalgarno sequence and that the "range" really just represents these two points. If so, the manuscript should say that explicitly and instead of claiming that the author's method matches the range say that it matches the two measurements, the one with and the one without a Shine-Dalgarno sequence.

4) It is quite disappointing that in spite of being an expert in modeling prokaryotic translation, the author consistently misspells Shine-Dalgarno in the entire manuscript!

5) Based on the response to the reviewers, please add "initial" in front of "secondary structure" in lines 307 and 311.

6) The author claims to have replaced all the upper case Ks used for rates by lower case ks. However, many of the captions to supplemental figures still contain upper case Ks for rates.

7) "then" -> "than" in line 462

Reviewer #2: My concerns were almost fully addressed, I have just two minor comments left:

1) I strongly encourage the author to publish the code on (e.g.) Github and state the link to this page in the paper (instead of the private one). Private homepages tend to disappear at some point and the reader has no chance to use the published code anymore. Also, there is no need to wait until the code is “ready” before publication on Github. In contrast, these repositories are meant for code “under construction” (version control with git etc).

2) “I am unaware of estimates available for adjusting the kinetic rates of the ribosome due to temperature changes”: A simple Arrhenius approach is probably sufficient to estimate rate differences between 37 and 42 degrees, similar to what has been done in PMID 25358034. Would that work in your case?

**Have the authors made all data and (if applicable) computational code underlying the findings in their manuscript fully available?**

Reviewer #1: Yes

Reviewer #2: Yes

PLOS authors have the option to publish the peer review history of their article (what does this mean?). If published, this will include your full peer review and any attached files.

Reviewer #1: No

Reviewer #2: No

Figure Files:

Data Requirements:

Reproducibility:

References:

---

## [Decision Letter · Decision Letter 2]

10 Jan 2023

Dear Dr. Dykeman,

We are pleased to inform you that your manuscript 'Modelling ribosome kinetics and translational control on dynamic mRNA' has been provisionally accepted for publication in PLOS Computational Biology.

Best regards,

Martin Meier-Schellersheim

Academic Editor

PLOS Computational Biology

Mark Alber

Section Editor

PLOS Computational Biology

Reviewer's Responses to Questions

**Comments to the Authors:**

Reviewer #1: The author has taken into account all the remaining issues and the manuscript is ready for publication now.

I am sorry if my expression of disappointment over the misspelling of an important term came across as not collegial. I do think that attention to detail matters, in presentation as well as in the research.

Reviewer #2: All my comments have been addressed, I have no further issues.

**Have the authors made all data and (if applicable) computational code underlying the findings in their manuscript fully available?**

Reviewer #1: Yes

Reviewer #2: Yes

PLOS authors have the option to publish the peer review history of their article (what does this mean?). If published, this will include your full peer review and any attached files.

Reviewer #1: No

Reviewer #2: No

---

## [Editor Report · Acceptance letter]

19 Jan 2023

PCOMPBIOL-D-22-01546R2 

Modelling ribosome kinetics and translational control on dynamic mRNA

Dear Dr Dykeman,

I am pleased to inform you that your manuscript has been formally accepted for publication in PLOS Computational Biology. Your manuscript is now with our production department and you will be notified of the publication date in due course.

With kind regards,

Anita Estes
